# The Trifecta: Three simple techniques for training deeper Forward-Forward networks

**Thomas Dooms**                                                                    *thomas.dooms@uantwerpen.be*
*Department of Computer Science*
*University of Antwerp*

**Ing Jyh Tsang**                                                                    *inton.tsang@imec.be*
*Department of Computer Science*
*University of Antwerp, imec*

**Jose Oramas**                                                                    *jose.oramas@uantwerpen.be*
*Department of Computer Science*
*University of Antwerp, sqIRL/IDLab, imec*

**Reviewed on OpenReview:** *https://openreview.net/forum?id=a7KP5uoOFp*

## Abstract

Massive backpropagated models can outperform humans on a variety of tasks but suffer from high power consumption and poor generalization. Local learning, which focuses on updating subsets of a model's parameters at a time, has emerged as a promising technique to address these issues. Recently, a novel local learning algorithm called Forward-Forward has received widespread attention due to its innovative approach to learning. Unfortunately, its application has been limited to smaller datasets due to scalability issues. To this end, we propose The Trifecta, a collection of three simple techniques that drastically improve the Forward-Forward algorithm on deeper networks. Our experiments demonstrate that our models are on par with similarly structured, backpropagation-based models in both training speed and test accuracy on simple datasets. Specifically, we achieve around 84% accuracy on CIFAR-10, a notable improvement (25%) over the original FF algorithm.

## 1 Introduction

Despite its pervasiveness, backpropagation is not flawless; it has inherent challenges that have puzzled researchers for decades. One such challenge is that it usually produces overfitted models, models that memorize the training set and fail to generalize to unseen data (Erhan et al., 2010). Over the years, researchers have found ways to mitigate the issue by introducing dropout (Srivastava et al., 2014) or simply increasing the dataset size through augmentation techniques or by collecting more data. However, with modern datasets reaching near-internet size (Gao et al., 2020; Schuhmann et al., 2022; Carlini et al., 2021), this provisional solution cannot be scaled further. Moreover, contemporary models are tricky to scale both locally (Nagel et al., 2021) due to memory issues and across machines (Huang et al., 2019) due to communication overhead. The need for sequential forward and backward passes leads to the phenomenon known as backward locking, hampering the efficient distribution of computation (Huo et al., 2018). Lastly, backpropagation struggles with vanishing or exploding gradients, which affect learning and stability, especially in deeper networks (Hochreiter & Schmidhuber, 1997; Glorot & Bengio, 2010; Bengio et al., 1994). Research has unveiled several mitigations (He et al., 2016; Ioffe & Szegedy, 2015; He et al., 2015; Glorot & Bengio, 2010), but this still remains an issue in large models (Chowdhery et al., 2022).

The combination of these systematic issues and the biological implausibility of backpropagation (Lillicrap et al., 2016) has prompted the machine learning community to explore alternative learning algorithms. One such approach is local learning (Bengio et al., 2006; Hinton et al., 2006; Belilovsky et al., 2019; Nø kland,

2016b), which learns from nearby information, instead of a full error signal. Limiting the information flow in the full network restricts the network's ability to memorize inputs and helps to regularize gradient computations. Further, the reduced communication facilitates the efficient deployment of these networks across several accelerator units (Gomez et al., 2022; Laskin et al., 2021).

Layer-wise or greedy learning is a popular family of local algorithms that aims to learn each layer in a neural network independently (Bengio et al., 2006; Belilovsky et al., 2019). It avoids backward locking during training and enables non-differentiable operations in the forward process. However, it often requires auxiliary classification heads or involves expensive mathematical operations (Belilovsky et al., 2019; Löwe et al., 2019; Nø kland, 2016b). Recently, a new local learning algorithm called Forward-Forward (FF) (Hinton, 2022) has sparked interest in the machine learning community. FF learns by maximizing or minimizing *goodness* from the activations of each layer for positive (genuine) and negative (bogus) samples, respectively. However, initial experiments show that this algorithm is unable to scale beyond the simplest of datasets. Therefore, in this paper, we aim to address this issue by pointing out three weak points of the algorithm and remedying them.

**Loss Function** In line with recent work (Lee & Song, 2023), we find that the proposed loss function from Hinton (2022) has several weaknesses. Primarily, due to asymmetries in the loss landscape, convergence is severely hindered. Moreover, the original loss function has a threshold parameter, which determines around which value of goodness the separation of positive and negative samples should occur. Our experiments show that this hyperparameter has a large impact on the final accuracy of the network, which is undesirable.

**Normalization Function** The objective function of each layer is to separate positive and negative samples as much as possible. This can cause an issue where layers can simply retain or scale prior features to achieve high separation without learning anything useful, as noted by Hinton (2022). Their proposed solution is to normalize the features length-wise at the start of each layer such that any information about the previous prediction is removed. This work confirms the occurrence of this issue but claims that the solution suffers from several flaws such as information loss and instability.

**Lack of Error Signals** A drawback of many greedy or local learning algorithms is the lack of backward or error signals. This means that a layer may learn features that are useful for its local objective but not necessarily for the next layer. Consider the extreme case where a layer generates a uniform feature vector ($k \cdot \mathbf{1}$) above or below the threshold according to the belief of the layer. Locally, the objective will be fulfilled, but subsequent layers will not acquire useful information that can help separate the data further. Consequently, the local objective must be suitable, or such models will not be able to scale to complex tasks.

## 1.1 Contributions

- We propose a combination of straightforward techniques, termed The Trifecta, that matches the accuracy of our backpropagation baseline on several datasets by addressing the mentioned weaknesses.

- We achieve around 84% accuracy on CIFAR-10 with a 12-layer model, demonstrating that FF can scale to more complex datasets and architectures.

- We perform an analysis on two different loss functions for FF and discuss their differences in training stability. This reveals that using a dynamic threshold significantly helps convergence in deeper networks.

- We conduct a thorough investigation into different normalization functions in the FF algorithm, which demonstrates that batchnorm strikes a good balance between feature recycling and information loss.

- We provide an implementation of The Trifecta (https://github.com/tdooms/trifecta).

## 2 Related Work

Before the advent of end-to-end backpropagation (E2EBP) (Krizhevsky et al., 2012), often simply called backpropagation (BP), the investigation into algorithms to train neural networks was prevalent in machine

learning research (Hinton & Sejnowski, 1983; Hinton et al., 2006). Since the backpropagation breakthrough, research has mostly shifted to improving this algorithm. Recently, however, there has been a resurgence of interest in alternate learning methods due to the limitations of backpropagation (Jaderberg et al., 2017; Nø kland, 2016b; Huo et al., 2018; Löwe et al., 2019). An early example of a localized approach in deep networks is the Inception Net (Szegedy et al., 2015), which incorporates multiple classification heads on intermediary layers throughout the network. Although the network still employs backpropagation, the intermediate heads aid gradient flow, enabling the training of deeper networks. Several years ago, a research wave focused on truly local algorithms began with the introduction of Decoupled Neural Interfaces (DNIs) (Jaderberg et al., 2017). DNIs employ independent sub-networks, called modules, allowing straightforward forward passes by sequentially passing inputs through all modules. The backward pass updates weights using synthetic gradients approximated using an auxiliary network trained offline with true gradients. This decoupling of gradient flow permits asynchronous forward and backward passes. However, the drawback of this approach is diminished accuracy, even on small datasets. More recently, Forward Gradient techniques have shown more promise. These methods calculate directional derivatives based on either random directions (Baydin et al., 2022) or directions informed by local auxiliary networks (Fournier et al., 2023). The resulting directional derivative serves as an estimate of the gradient along the chosen direction. Concurrently, increasingly biologically inspired methods have emerged, starting with feedback alignment (FA) (Lillicrap et al., 2016) that proposes to use random feedback matrices to resolve the implausibility of synaptic symmetry within backpropagation. Nø kland (2016a) alters this approach by using random matrices to directly update each layer, partly resolving the backward locking issue. Lastly, PEPITA (Dellaferrera & Kreiman, 2022) uses two forward passes to learn: a standard pass and a modulated pass. The first pass is comparable to an ordinary forward pass in backpropagation and computes the network error for a given input. The original input is then perturbed by this error and used by the second pass to update the weight directly in a forward manner.

The Forward-Forward algorithm (Hinton, 2022) has been proposed as a novel approach to deep learning. The FF algorithm aims to overcome several limitations tied to backpropagation and provide a biologically plausible learning algorithm. This is achieved by adopting contrastive learning principles to separate positive and negative samples based on network activations. The lack of error signals and the generic objective make the algorithm very flexible in many scenarios where backpropagation is not feasible. Unfortunately, this approach often results in reduced accuracy on complex datasets. Some work has been done to improve the scaling characteristics of the FF algorithm. Specifically, the original loss function exhibits conditioning issues due to its suboptimal loss landscape, resulting in difficult optimization. To mitigate this, a modified loss function known as SymBa directly utilizes the separation instead of the distance to a threshold. This leads to improved properties and significantly enhanced performance (Lee & Song, 2023).

This paper explores and improves the Forward-Forward algorithm in an image classification context. We achieve this by proposing three minor modifications to the algorithm, including using the SymBa loss function (Lee & Song, 2023), batch normalization (Ioffe & Szegedy, 2015), and overlapped local updates (Laskin et al., 2021). To the best of our knowledge, we are the first to investigate the effect of the latter two in a Forward-Forward context. Further, we do not specifically focus on the biological plausibility but attempt to stay as close to the nature of the algorithm as possible. For instance, we employ convolutional neural networks (CNNs) (LeCun et al., 1989) due to their ubiquity in image processing, even though weight sharing is not biologically plausible (Grossberg, 1987).

## 3    Background

The Forward-Forward algorithm is a biologically plausible alternative to backpropagation. Instead of minimizing the cross-entropy between the model's outputs and a one-hot encoding of labels, it replaces the backward pass from backpropagation with a second forward pass, hence the name. The first forward pass contains positive (real) data, and the second pass contains negative (bogus) data. The main idea is that each layer of the network should learn to distinguish this data as well as possible. To this end, Hinton (2022) considers the magnitude of hidden activations, which they call *goodness*. This goodness should be high for positive samples and low for negative samples. Generally, the $l_2$ norm is used to determine the goodness, but variants exist.

Importantly, the neural network structure and design remain the same; each layer modifies the incoming features according to their weights and forwards the result to the next layer. Intuitively, instead of using a gradient of the global loss (computed at the output layer) with respect to the weights, a proxy loss (the goodness) is computed at each layer and applied locally, obviating the need for backpropagation. This brings about three differences compared to ordinary supervised training, which we discuss in detail in this section.

### 3.1 Negative Sampling

At the core of the learning algorithm lies learning the distinction between positive and negative samples. Consequently, the nature and quality of negative samples strongly influence the learning process. While one can imagine a multitude of ways to approach this, we focus on creating negative and positive samples in a supervised manner. Specifically by adding the label to the input. In this case, a positive sample is a pair in which the input and label match. Conversely, a negative sample has labels and inputs that don't match. To distinguish matching pairs, the model has to learn the structure of the inputs. In some cases, one should be careful about how this pair is created. Consider a CNN-based model where the label is encoded in a limited number of pixels; then, not all parts of the model could access this data. Due to the lack of backward signal, this may seriously hamper learning. While such a contrastive approach has some drawbacks, often rooted in lower computational efficiency, it is much more flexible. For instance, one could focus on the negative pairs that the model often mispredicts or are important to the task.

### 3.2 Weight Updates

Rather than performing a forward prediction step followed by a backward update step, two forward prediction steps are performed on positive and negative data. Each layer updates itself locally (without receiving gradients) based on the generated goodness with the following loss function.

$$L_{FF}(g^{pos}, g^{neg}) = \sum_i log(1 + exp(\tau - g_i^{pos})) + \sum_i log(1 + exp(g_i^{neg} - \tau))$$

Here, $g^{pos}$ and $g^{neg}$ refer to the vector of the goodness values of the positive and negative samples in each batch, respectively, and $\tau$ is a threshold. This threshold defines how strongly the positive and negative samples should be separated. Finally, $i$ represents an entry in a batch. This function will produce low loss values if both the goodness of positive samples is well above the threshold and well below the same threshold for negative samples.

To update each layer's weights, the optimizer step is performed using the gradient of this loss function using a standard optimizer [1]. The update rule for SGD, for example, would be:

$$W_{t+1} = W_t - \eta \nabla L_{FF}(g^{pos}, g^{neg})$$

where $W_t$ represents the weights at time step $t$, $\eta$ is the learning rate, and $\nabla L_{FF}(g^{pos}, g^{neg})$ is the gradient of the loss function with respect to the weights. We illustrate this algorithmically in 1.

---

**Algorithm 1** Original Layer Update Step

---

1: $\text{hidden}_{i+1} \leftarrow \text{layer}_i(\text{hidden}_i)$
2: $\text{activations} \leftarrow \text{out.pow(2).flatten(start\_dim=1)}$
3: $\text{pos, neg} \leftarrow \text{activations.chunk(2)}$
4: $\text{loss} \leftarrow \text{softplus}(\tau - pos).\text{mean()} + \text{softplus}(neg - \tau).\text{mean()}$
5: $\text{loss.optimize()}$
6: **return** $\text{detach(hidden}_{i+1})$

---

[1]Alternatively, since the update step only uses local weights, one could use exact methods on small datasets.

### 3.3 Evaluation

During evaluation, the goodness across one or multiple layers determines the certainty in the prediction of the provided sample. In the supervised setting, a forward pass with each label must be performed to attain the logits of the n-way classification. This can be inefficient, especially in scenarios with numerous classes, such as ImageNet (Deng et al., 2009). Alternatively, the features of the last layer in the network can be used to train a classifier; this mitigates the linear complexity of n-way classification [2]. Due to the novelty of the algorithm, different existing training and evaluation strategies are largely unexplored. Consequently, FF often exhibits lower accuracy and performance compared to other local learning techniques. Nevertheless, this presents a promising avenue for investigating learning algorithms and gaining insights into the learning process of deep networks in general. Appendix H discusses some experiments and ideas in this area.

This work focuses on the simplest version of the Forward-Forward algorithm. We encourage the reader to refer to the original work (Hinton, 2022) for a more complete intuition and connections to other learning algorithms.

## 4 The Trifecta

The main contribution of this work is The Trifecta, which aims to solve three observed weaknesses in the Forward-Forward algorithm: the loss function, the normalization, and the lack of error signals. As our experimentation reveals, there is a positive synergy between the three components; combining the three yields higher gains than their separate improvements.

### 4.1 Symmetric Loss Function

Consistent with (Lee & Song, 2023), we find that the proposed loss function in FF is not equal for false positives and false negatives of the same scale, leading to gradients that do not directly point toward the minimum loss. Additionally, it is not possible to have negative goodness which means the separation of the negative goodness is limited for many threshold values. The original paper (Hinton, 2022) noted that minimizing goodness, instead of maximizing it for positive samples, yields higher accuracy. This suggests that it is easier for the network to distinguish negative examples very well according to how bogus they are. Appendix C provides a comprehensive mathematical analysis of these issues. To address these issues Lee & Song (2023) proposes the SymBa loss function that relies on directly optimizing for the separation gap, shown below.

$$L_{SymBa}(g^{pos}, g^{neg}) = \sum_i log(1 + exp(g_i^{pos} - g_i^{neg}))$$

This resolves the aforementioned issue. This approach works best when the positive and negative samples are highly correlated, which is the case with the supervised approach. Additionally, this function can be improved by scaling the separation with a constant term $\alpha$, this penalizes wrong classifications further, akin to focal loss (Lin et al., 2018). Our work, therefore, opts to employ this loss function. In Appendix D we further discuss the advantages and disadvantages of this loss function.

### 4.2 Batch Normalization

Prior to each layer, Hinton (2022) suggests normalizing the length of the activations, akin to a simplified form of layer normalization (LN) [3]. This serves two purposes: it prevents prediction information from leaking

---

[2]In the supervised approach, a label must still be provided during evaluation. The original work (Hinton, 2022) uses a 'neutral' label which is an average of all labels. Limited experiments show that simply using a random label works slightly better in practice.

[3]In this work we refer to this reduced form of normalization that does not subtract the mean simply as layernorm. All experiments in this regard are verified to hold for both this reduced version and the standard PyTorch layernorm implementation (PyTorch Contributors, 2023).

into the next layer and forces the network to learn completely new features at each layer. By scaling the network depth-wise, each layer has an alternate insight into the data which is used by the final prediction.

However, we find this not to be the case in the original version of FF; deeper layers do not produce features that lead to higher accuracy. Instead, the accuracy plateaus by scaling the depth (Figure 1). We argue that this stems from the combination of the normalization and the lack of error signals. This disruptive normalization that removes all previous prediction information prohibits the network from creating incrementally useful and specific features observed in hierarchical learning in backpropagated networks. In combination with the fact that there are no global error signals, the network is unable to optimize across layer boundaries and therefore cannot improve downstream accuracy. In essence, each layer has to fully re-separate the provided features, leading to a collection of single-layer classifiers with shallow insight. This stack of classifiers operating on the output of their predecessor leads to high instability, especially at later layers, shown in Figure 2.

Intuitively, we seek a transformation that strikes the balance between doing nothing, which leads to recycling features, and discarding all prediction information which leads to each layer having to re-separate all features. To this end, we propose to use batch normalization (BN) (Ioffe & Szegedy, 2015). Batch Normalization normalizes features within each batch without removing the information regarding the previous prediction. Further, as this normalization is dependent on batch statistics, which are not available to any single sample, any subsequent layer cannot exactly reconstruct the original features. Additionally, directly using batch-normalized features most likely will not lead to high goodness, especially if these lead to a wrong prediction. This forces new features or insights to be created at each layer, which is empirically demonstrated on the bottom of Figure 1. In essence, each layer has the ability to alter features instead of separating them again. This leads to lower weights Figure 10, drastically stabler gradients Figure 2, and therefore better generalization. This is discussed more rigorously and intuitively in Appendix E.

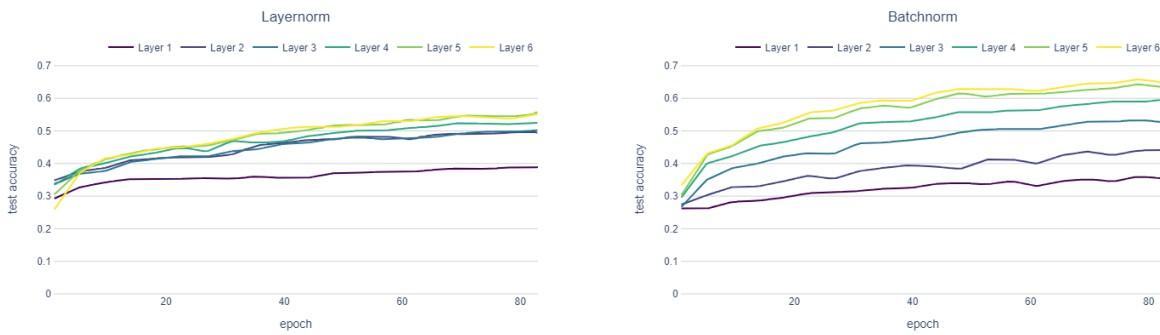

Figure 1: Layerwise test accuracy for both normalization schemes. Both experiments use SymBa, but not overlapped local updates on the shallow CNN network (VFF+SymBa/s) on CIFAR-10.

## 4.3 Overlapping Local Updates

As a means to introduce error signals into FF, the original work (Hinton, 2022) proposes a recurrent network that uses the goodness of the subsequent and previous layers. However, for simple classification, this requires the network to be evaluated several times. Therefore, we propose using another technique from recent research (Huo et al., 2018; Laskin et al., 2021) that uses semi-local error signals within the context of backpropagation. This approach involves training layers in groups in an alternating and overlapping pattern. Using group size 1 simply results in greedy learning while using group size 2 leads to Overlapping Local Updates (OLU). A more thorough explanation and further implementation details can be found in Appendix F.

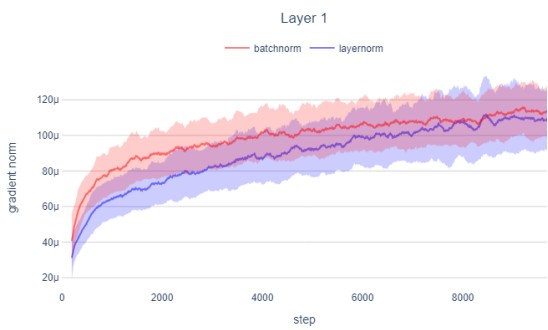 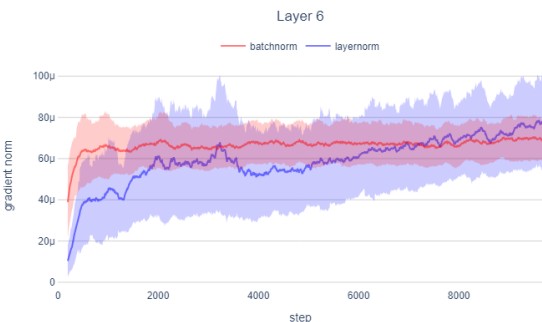

Figure 2: $\ell_2$ norm of the gradients of layers 1 and 6 at each step during training using the FCN network (VFF+SymBa/f) on CIFAR-10. OLU is not used due to its potential influence on the results. Using batchnorm yields more stable gradients in deeper layers.

In essence, this technique optimizes each layer with two alternating objectives. When the layer is last in the group, it is updated to satisfy the local objective: maximizing goodness. Conversely, when it is first in the group it is updated to better satisfy the objective of the subsequent layer: help the subsequent layer maximize goodness. Intuitively, the former promotes local understanding of the data and the latter prompts the layer to increase the usefulness of its representations for the next layer. In essence, in accordance with the original idea of Hinton (2022), local weights are updated according to the beliefs of the subsequent layer. Many alternate techniques exist to incorporate error signals into local networks (Ororbia & Mali, 2023; Lillicrap et al., 2016) in a more biologically plausible manner, however, exploring these is deferred to future work.

In summary, these three changes result in the algorithm (shown in 2) to update a layer.

---

**Algorithm 2** Modified Layer Update Step

---

1: $\text{hidden}_{i+1} \leftarrow \text{layer}_i(\text{hidden}_i)$
2: **if** $(i \bmod 2 \equiv 0) \oplus (\text{iteration} \bmod 2 \equiv 0)$ **then**
3:     **return** $\text{hidden}_{i+1}$
4: **end if**
5: $\text{activations} \leftarrow \text{out.pow(2).flatten(start\_dim=1)}$
6: $\text{pos, neg} \leftarrow \text{activations.chunk(2)}$
7: $\delta \leftarrow \text{neg.mean(1) - pos.mean(1)}$
8: $\text{loss} \leftarrow \text{softplus}(\alpha \cdot \delta).\text{mean}()$
9: $\text{loss.optimize}()$
10: **return** $\text{detach}(\text{hidden}_{i+1})$

---

## 5 Experimental Setup

Conforming to Hinton (2022), all experimentation in this work concentrates on image classification to measure the capabilities of the Forward-Forward algorithm, specifically on the MNIST, Fashion-MNIST, SVHN, CIFAR-10, and CIFAR-100 datasets (LeCun & Cortes, 2010; Xiao et al., 2017; Netzer et al., 2011; Krizhevsky, 2009). These datasets provide a good difficulty mix to assess the behavior of the algorithm in different scenarios.

### 5.1 Training and Encoding

In this work, we opt to exclusively employ the supervised approach to the Forward-Forward algorithm as described in Hinton (2022). Further, we opt to use CNNs due to their ubiquitous nature in image classification. This combination poses some problems as the one-hot label encoding (1D) cannot be trivially concatenated to an image (3D). Additionally, the encoding must be concatenated in such a way that a majority of convolutions can differentiate the labels [4]. To this end, this work encodes the label into an additional channel of identical size as the images, this encoding is achieved through a learned matrix in the case of OLU and a random one without. In the case of CIFAR-10, the one-hot vector is multiplied by a $10 \times 1024$ embedding matrix ($E$), and this result is transformed into a $32 \times 32$ matrix and concatenated as a fourth input channel (shown below). In practice, we found this to work well.

$$g_i^{pos} = [x_i, \text{reshape}(Ey_i, 32, 32)]$$
$$g_i^{neg} = [x_i, \text{reshape}(E\bar{y}_i, 32, 32)]$$

Where $[\cdot, \cdot]$ defines channel-wise concatenation, $y_i$ is the label corresponding to $x_i$ and $\bar{y}_i$ is a wrong label.

### 5.2 Architecture

Selecting the appropriate architecture is a crucial consideration in deep learning. However, most design principles and architectural rules-of-thumb stem from backpropagation and are not blindly applicable to unexplored learning algorithms. For instance, in a local learning setting, bottleneck layers most often result in information loss rather than compact representations in our experience. Currently, the Forward-Forward algorithm has only been used to train small fully-connected networks and non-weight-sharing local receptive fields. In line with other work within local learning (Nøkland, 2016b; Belilovsky et al., 2019), this work employs VGG-like CNNs (Simonyan & Zisserman, 2015). However, some specific changes are made to adapt it to Forward-Forward. First, we omit the (linear) classification head because goodness is simply determined from an aggregation of activations, no matter the dimensionality [5]. Second, we use a different ordering of operations within each layer. Specifically, we first perform normalization, followed by the convolution, the non-linearity, and finally an optional maxpool. This is simply an extension of Hinton (2022) and permutations of this have not been explored further in this work [6]. This work studies three architectures: two CNNs consisting of 6 and 12 layers and an FCN with 6 layers. These networks respectively consist of 2.8 million, 8.5 million, and 25.1 million parameters (depending on the input). The CNN architectures are depicted in Figure 12. The FCN is simply a stack of hidden layers with a dimension of 2048.

### 5.3 Evaluation Metrics

The Forward-Forward algorithm allows for more fine-grained metrics to evaluate any given network compared to backpropagation. In essence, each layer has the same objective, in contrast to backpropagation where only the last layer has an objective, which can be used to examine the relations between layers. For instance, it is possible to inspect how well a single layer is able to classify based solely on its own goodness. We expect this accuracy to increase with depth, and failure to do so may indicate an issue with the previous layers. The same principle can be applied to any per-layer metric, such as the separation, loss, or gradient stability to gain insight into the behavior of the model. An important metric in evaluating learning algorithms is training accuracy. This indicates whether the algorithm is overfitting or simply fails to represent the dataset. However, in the supervised setting, this is impossible to determine efficiently without sampling all labels, which would slow down training time by the number of labels.

---

[4]Consider the case where the first ten pixels of the image are used to encode the label. The majority of output neurons have no information about the label, leaving them incapable of separating positive from negative.

[5]Preliminary experiments in this work have shown that the goodness mechanism, in general, does not operate well when using very few features, such as in a classification head.

[6]For simplicity, we do not apply any techniques such as the peer-normalization proposed in Hinton (2022) in all networks.

### 5.4 Accuracy

To measure the actual accuracy of the network, Hinton (2022) reports that taking the average goodness of all layers except the first leads to the highest accuracy. Our experiments show this to be true in tiny networks trained with the original algorithm but when using The Trifecta on deeper networks, accuracy drastically degrades. This is because, on complex datasets, the earlier layers are significantly worse than the later ones and affect the stability of the ensemble. Therefore, all reported single-valued accuracies (Table 1, Table 2) are simply based on the goodness of the last layer. Nevertheless, in deep networks, aggregating the last few layers can be fruitful. By aggregating the last 3 layers for goodness-based classification, we can improve test accuracy by a few decimal points. More information about evaluation can be found in Appendix H.

## 6 Results

We perform and study two series of experiments to evaluate the effectiveness of The Trifecta. The first is an ablation of the three components of The Trifecta. The second is a study into the impact of the architecture and training time. Where possible, we compare results to the vanilla (original) Forward-Forward algorithm (Hinton, 2022), feedback alignment (Lillicrap et al., 2016), PEPITA (Dellaferrera & Kreiman, 2022), and a backpropagation baseline. This baseline is trained on the deep architecture with an altered layer structure to accommodate backpropagation [7]. Further, a linear classification head is appended. Note that this simply serves as a baseline and is far from SOTA. The same hyperparameters and architecture are used on all datasets for fair comparison. We describe each model with the shorthand notation [algorithm]/[architecture]. The considered algorithms consist of Vanilla FF (VFF), Trifecta FF (TFF), feedback alignment (FA), PEPITA (PEP), and backpropagation (BP). The architectures are the shallow CNN (s), the deep CNN (d), the FCN (f), and finally a tiny network consisting of 3 or fewer layers (t). For instance, VFF/s denotes the original FF algorithm on the shallow architecture.

### 6.1 Ablation of Trifecta Components

Table 1 shows the accuracy achieved by multiple interplays of The Trifecta components. We perform this ablation with the shallow architecture on the CIFAR-10 dataset as it is the most challenging and, therefore, differentiates the algorithms best. The first row illustrates the importance of the loss function in our recreation of the original FF algorithm. This highlights the importance of $\tau$ as a hyperparameter. The value $\tau = 2$ is chosen in line with other implementations (Lee & Song, 2023; Pezeshki, 2023) and $\tau = 10$ is an optimized value we found for our network on CIFAR-10. Additionally, utilizing the improved loss function further improves the accuracy and establishes this component as the most crucial part of the Trifecta. The next rows show that the addition of each component yields significant gains to the final accuracy. The full Trifecta (the bottom right entry) achieves the highest accuracy by a significant margin. The last column shows that the combination of all techniques outperforms the sum of parts, the differences compared to only SymBa are 6.60% for OLU, 9.29% for BN, and their combination is 16.01%.

### 6.2 Convergence Properties

We train the TFF/s and the TFF/d architectures for 200 and 500 iterations respectively on four image classification datasets (Table 2). We compare our results at several stages of training to the available results from Hinton (2022) and a backpropagation baseline. When using backpropagation to fit trivial datasets, such as MNIST, unnecessarily deep networks are inadvisable as they lead to slower convergence and instability. Our experiments confirm this observation as the deep model (99.58%) only slightly outperforms the shallow model (99.53%). A similar trend can be observed for the f-MNIST and SVHN datasets. On the other hand, the more challenging CIFAR-10 dataset exhibits the same behavior up to 100 epochs, where the shallow model (75.23%) convincingly outperforms the deep model (71.12%). However, the longer training time allows the deep model (83.51%) to surpass the shallow model (80.01%). A more detailed explanation of this phenomenon can be found in Appendix G. Lastly, the learning speed of our algorithm is very competitive,

---

[7]Backpropagation, trained on the shallow architecture, shows slightly worse convergence properties and is therefore not shown. Further, on our setup, backpropagation converges after 100 epochs, showing no further improvement on longer training.

which can be seen on the row depicting accuracy after 10 epochs. These results highlight the potential of The Trifecta on the Forward-Forward algorithm. Three slight changes have drastically improved the accuracies of the learning algorithm for all datasets: on simple datasets, our algorithm is only slightly behind our backpropagation baseline, and on more complex datasets, we obtain very competitive accuracies compared to other approaches.

Table 1: Test accuracy ablation with only specific components of The Trifecta using the shallow network on CIFAR-10 after 100 epochs over 3 runs. Referring to the original loss, the shorthand FTL (Fixed Threshold Loss) is used in this table.

|  | FTL ($\tau$=2) | FTL ($\tau$=10) | SymBa ($\alpha$=4) |
|---|---|---|---|
| None | 29.27 $\pm$ 2.49 | 44.28 $\pm$ 1.28 | 59.22 $\pm$ 0.49 |
| BN | 41.02 $\pm$ 1.89 | 37.31 $\pm$ 3.01 | 68.51 $\pm$ 0.27 |
| OLU | 48.18 $\pm$ 1.87 | 50.85 $\pm$ 0.63 | 65.82 $\pm$ 0.78 |
| OLU+BN | 57.92 $\pm$ 1.11 | 58.99 $\pm$ 0.84 | **75.23** $\pm$ 0.65 |

# 7 Discussion and Future Work

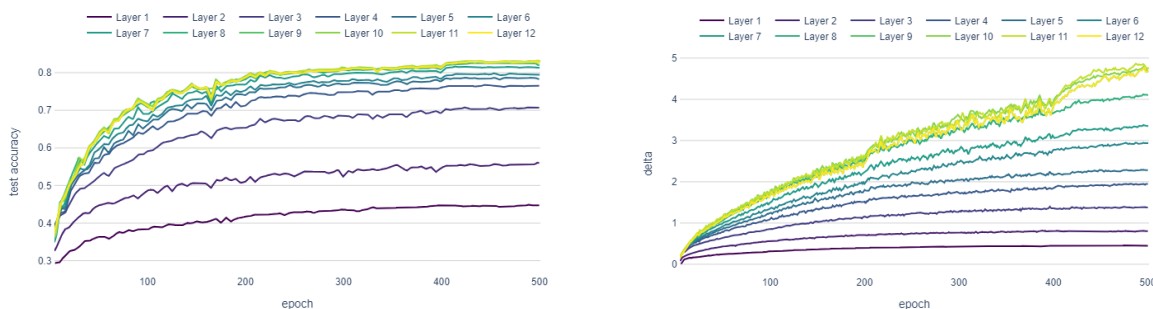

Figure 3: The evolution of test accuracy (left) and separation (right) throughout the training of the best-performing CIFAR-10 TFF/d model presented in Table 2.

We can split the empirical impact of The Trifecta on the Forward-Forward algorithm into two categories. First, the progressive improvement is the amount by which each layer improves compared to its predecessor. Ideally, appending additional layers continues to improve accuracy. Second, the learning speed is the amount by which each layer improves over time. Trivially, the improvement of each layer must be as quick as possible to reduce training time.

## 7.1 Progressive Improvement

The primary advantage of The Trifecta lies in its ability to improve the accuracy upon going deeper into the network. In contrast to the original FF, which is unable to learn further than a few layers, especially when using CNNs (Figure 4). In this work, we were able to scale to 12 layers on multiple datasets with considerable gains per layer. Above this threshold, our experiments did not show worthwhile improvement. Further, the progressive improvement of later layers is closely related to the learning rate. Figure 3 illustrates the accuracy and the separation throughout training. These plots reveal the impact of a lower learning rate on the separation and stability of the later layers, as can be seen at epochs 200 and 400 when the learning rate schedule changes. This is also reflected in the accuracy (although difficult to see on the plot), as the difference between layers 6 and 12 at epoch 200 is 2.6% and 3.7% at epoch 500. This work has only scratched the surface in terms of exploring the impact of learning rate schedules. More exotic learning schedules, which may involve freezing certain layers, may be highly fruitful to further improve accuracy.

Table 2: Test accuracy comparison between learning algorithms on several datasets at certain epochs. We show the mean and standard deviation of 3 runs. Note that these are the accuracies for the last layer alone, we achieve our best results by combining the last three layers as outlined in Appendix H. The VFF/t and PEP/t results are from Hinton (2022) and Dellaferrera & Kreiman (2022) respectively.

| model | epochs | MNIST | F-MNIST | SVHN | CIFAR-10 | CIFAR-100 |
|-------|--------|-------|---------|------|----------|-----------|
| FA/f | 100 | $98.60 \pm 0.08$ | $85.87 \pm 0.19$ | $71.11 \pm 0.66$ | $53.39 \pm 0.24$ | $23.00 \pm 0.28$ |
| PEP/t | 100 | $98.29 \pm 0.13$ | | | $56.33 \pm 1.35$ | $27.56 \pm 0.60$ |
| VFF/t | 60 | $98.6$ | | | | |
| VFF/t | 500 | $99.4$ | | | $59$ | |
| TFF/s | 10 | $97.44 \pm 0.49$ | $76.81 \pm 2.37$ | $84.62 \pm 1.34$ | $50.32 \pm 0.32$ | $11.32 \pm 1.57$ |
| TFF/s | 100 | $99.23 \pm 0.11$ | $88.00 \pm 0.67$ | $92.57 \pm 0.67$ | $75.23 \pm 0.65$ | $28.71 \pm 0.42$ |
| TFF/s | 200 | $99.53 \pm 0.04$ | $91.38 \pm 0.38$ | $94.25 \pm 0.54$ | $80.01 \pm 0.60$ | $35.79 \pm 0.17$ |
| TFF/d | 100 | $99.32 \pm 0.11$ | $85.62 \pm 0.53$ | $91.49 \pm 0.34$ | $71.12 \pm 0.77$ | $24.42 \pm 0.55$ |
| TFF/d | 200 | $99.28 \pm 0.03$ | $87.90 \pm 0.42$ | $92.60 \pm 0.27$ | $78.99 \pm 1.16$ | $28.88 \pm 0.80$ |
| TFF/d | 500 | $\mathbf{99.58} \pm 0.06$ | $91.44 \pm 0.49$ | $\mathbf{94.31} \pm 0.07$ | $83.51 \pm 0.78$ | $35.26 \pm 0.23$ |
| BP/d | 100 | $99.07 \pm 0.10$ | $\mathbf{93.61} \pm 0.11$ | $94.01 \pm 0.20$ | $\mathbf{89.32} \pm 0.45$ | $\mathbf{59.64} \pm 0.46$ |

## 7.2 Learning Speed

Another strength of The Trifecta is its ability to learn more quickly. Each component in The Trifecta not only impacts the final accuracy of the model but also the speed at which this accuracy is achieved. All components contribute to this in their own way, leading to the strong synergy within The Trifecta. Specifically, our shallow CNN model is able to cross the cape of 97% on MNIST and 50% on CIFAR-10 in less than 10 epochs. Again, the learning speed is closely related to the learning rate of the layers. We are confident that finding techniques to increase the learning rate, will further improve the speed of FF.

The focus of this work is to improve the accuracy of the FF algorithm in a supervised image classification setting using slight modifications. However, outside of The Trifecta and the original Forward-Forward paper (Hinton, 2022), not much has yet been explored and several ideas for improving the algorithm still remain untried and await further investigation. We strongly believe that future work in the following two areas is most important for the future of FF: finding better evaluation strategies and improving general understanding of the learning characteristics.

## 7.3 Evaluation Strategies.

Ultimately, FF is still held back due to several issues. In particular, performing non-binary classification using this algorithm is prohibitively slow as each class needs to be sampled separately to achieve high accuracy. Using FF for tasks with many classes, such as classifying ImageNet (Deng et al., 2009), is therefore strongly inadvisable. Nonetheless, alternate classification techniques that avoid sampling all classes separately by using candidate lists or one-pass classification (Hinton, 2022) seem promising but currently lead to lower accuracy and higher instability, respectively. Solving this would be a huge boost toward the viability of Forward-Forward in more tasks.

## 7.4 General Understanding.

There are several gaps in knowledge about FF which we believe to be nontransferable from backpropagation. For instance, performing a thorough architecture search to determine which patterns and blocks work best. Bottleneck layers are not suitable for Forward-Forward but inverted bottlenecks (Sandler et al., 2018) may be. Additionally, techniques such as residual connections (He et al., 2016) remain an intriguing avenue of research that may push the algorithm to scale to increasingly complex tasks (touched upon in Appendix I). Further, an investigation using well-established visual explanation techniques such as (Selvaraju et al., 2019)

is sure to reveal interesting properties about FF. Interestingly, these methods can be performed layer-wise to ascertain additional insight into the model behavior.

## 8 Closing Remarks

We proposed The Trifecta, three simple techniques that improve the accuracy and learning speed of FF-based methods. This enables the training of deeper FF-based architectures that are competitive with their backpropagated counterparts. We note that, within our proposed improvements, there remain several open questions. We encourage further research to scrutinize our findings as well as explore how to further improve the FF algorithm. Continued research into alternate learning algorithms may reveal generally applicable knowledge that will aid our understanding of machine learning and intelligence itself.

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

## A    Notation

Throughout the appendices, some notational shorthands are used that will be outlined here. Additionally, for simplicity, we will be analyzing the case of a single positive and negative example, instead of full batches.

First, we define a shorthand for the main component of the loss function, which is called softplus.

$$\xi(x) = softplus(x) = log(1 + exp(x))$$

Many functions can be used to calculate goodness, in line with the original work, we stick with the mean of squares, which can also be denoted as the square $\ell_2$ norm. *positive* and *negative* refer to the feature vectors from a positive and negative sample respectively.

$$g^{pos} = ||positive||_2^2 \quad g^{neg} = ||negative||_2^2$$

Often, we wish to examine the separation to the threshold, both for positive and negative examples.

$$\delta^{pos} = g^{pos} - \tau \quad \delta^{neg} = \tau - g^{neg}$$

In the same spirit, the overall separation between positive and negative is also useful.

$$\delta = \delta^{pos} + \delta^{neg} = g^{pos} - g^{neg}$$

The following is a formulaic summary of the three main components of the paper.

SymBA: $L_{SB} = softplus(\alpha \cdot \delta)$ where $\alpha$ is a hyperparameter that defines how strongly the model is punished for wrong predictions.

OLU: $(layer \bmod 2 \equiv 0) \oplus (iteration \bmod 2 \equiv 0)$ where $\oplus$ refers to a xor operation. This determines when the layer is updated in the update process as seen in Figure 11.

BN: $\left(x^{(k)} - E(x^{(k)})/Var(x^{(k)})\right) \cdot \gamma + \beta$ where the mean and variance are taken over the batch dimension.

## B  Hyperparameter Discussion.

Similar to backpropagation, training deeper networks requires more epochs. Specifically, we opted for 200 epochs for the shallow network and 500 for the deep network. Further, in line with Pezeshki (2023); Lowe (2023), we found the Forward-Forward algorithm is quite sensitive to the learning rate. Specifically, higher learning rates tend to lead to instability, and lower learning rates simply lead to slow convergence. We have found $10^{-3}$ to be a sensible default. For optimal results, however, we found a learning rate schedule to be highly beneficial. In our experiments, we train the deep network with a learning rate of $10^{-3}$ for the first 200 epochs, then $5 \cdot 10^{-4}$ for the next 200 epochs and $10^{-4}$ for the last 100. The shallow network uses $10^{-3}$ for the first 150 epochs and $10^{-4}$ during the last 50. In terms of batch size, we found 1024 (512 positive and 512 negative) to work well but changing this slightly does not seem to noticeably influence accuracy. Further, both discussed loss functions have a hyperparameter. For the original loss function, we mention the value of $\tau$ in each experiment explicitly. The SymBa loss has a scaling parameter that penalizes incorrect classification, we use a scale of 4 as recommended by Lee & Song (2023). We find that values in the range $[2; 8]$ for this parameter do not impact accuracy, only the separation. In terms of data augmentation, a random rotation and random crop are performed. Lastly, all networks are initialized using uniform Kaiming initialization He et al. (2015).

## C  Issues With the Original Loss Function

In this section of the appendix, following the previously introduced notation, we more formally indicate the issues with the original loss function. For convenience, the original loss function is repeated.

$$L_{FF}(g^{pos}, g^{neg}) \triangleq \xi(\tau - g^{pos}) + \xi(g^{neg} - \tau) \tag{1}$$

This study inspects whether the loss function handles positive and negative samples equally. As stated earlier, in reality, the loss function is computed for a batch of positive and negative samples (shown in section 3) but we will use $g^{pos}$ and $g^{neg}$ as the goodness of a single positive and negative sample respectively.

**1. Symmetry**: False positives and false negatives have an equal loss.

$$L_{FF}(\tau + \delta^{pos}, \tau + \delta^{neg}) \stackrel{?}{=} L_{FF}(\tau - \delta^{pos}, \tau - \delta^{neg}) \tag{2}$$

Intuitively, the left-hand side corresponds to the loss of correctly classifying a positive sample and wrongly classifying the negative one (false positive). The right-hand side represents the inverse, where the positive sample is misclassified (false negative). As the respective errors are the same, we expect the loss function to be the same. When filling in Equation 2 with Equation 1, we attain the following.

$$\xi(\tau - (\tau + \delta^{pos})) + \xi((\tau + \delta^{neg}) - \tau) = \xi(\tau - (\tau - \delta^{pos})) + \xi((\tau - \delta^{neg}) - \tau)) \tag{3}$$

$$\xi(-\delta^{pos}) + \xi(\delta^{neg}) = \xi(\delta^{pos}) + \xi(-\delta^{neg}) \tag{4}$$

We can see that this is the case iff $\delta^{pos} = \delta^{neg}$. i.e. when positive and negative samples are equally separated from the threshold.

**2. Equal Gradients**: Negative and positive samples are equally pushed away from the threshold.

$$\frac{\partial}{\partial \delta^{neg}} L_{FF}(\tau + \delta^{pos}, \tau + \delta^{neg}) \overset{?}{=} \frac{\partial}{\partial \delta^{pos}} L_{FF}(\tau - \delta^{pos}, \tau - \delta^{neg}) \tag{5}$$

In essence, the first part computes what the partial derivative of the negative goodness is upon a false positive. The second computes the same but for the positive goodness upon a false negative. If any one of these is misclassified, derivatives of equal value are desired. To verify whether this holds for the loss function, the partial derivatives need to be computed first.

$$\frac{\partial}{\partial \delta^{neg}} \xi((\tau + \delta^{neg}) - \tau) = \frac{\partial}{\partial \delta^{pos}} \xi(\tau - (\tau - \delta^{pos})) \tag{6}$$

$$\frac{\partial}{\partial \delta^{neg}} \xi(\delta^{neg}) = \frac{\partial}{\partial \delta^{pos}} \xi(\delta^{pos}) \tag{7}$$

$$\frac{exp(\delta^{neg})}{exp(\delta^{neg}) + 1} = \frac{exp(\delta^{pos})}{exp(\delta^{pos}) + 1} \tag{8}$$

Again, this is the case iff $\delta^{pos} = \delta^{neg}$, when positive and negative samples are equally separated from the threshold.

**Consequences.** The loss function only satisfies our two criteria given a very specific condition: when $\delta^{pos} = \delta^{neg}$. Intuitively, one would expect this to hold. However, if $\delta^{pos}$ increases, so does, $exp(\delta^{pos})/1 + exp(\delta^{pos})$ and similarly for $\delta^{neg}$, especially for small $\delta$, which occurs most of the time. Therefore, if the separation of either positive or negative increases, that derivative will start to dominate the gradient, leading to an unstable equilibrium.

Such imbalance can occur simply due to the variance in batches or due to it being simpler to generate high or low goodness for the optimizer. Another factor that can cause this imbalance is that goodness must be positive. This sets a hard limit for the separation of negative samples, namely $\delta^{neg} \leq \tau$. In reality, this limit is never even reached due to the optimization process. Achieving low goodness values necessitates a delicate equilibrium, rendering the network highly responsive to any minor alteration. Since the loss function's constituent terms are aggregated, the gradients stemming from positive samples severely disturb this balance. Conversely, this is not an issue for the positive samples as small disturbances from the negative gradients will not impact them as much. Scaling the threshold to be high enough to circumvent this behavior is not possible without repercussions; a high threshold will induce lots of instability in the network as all weights need to be higher, in some cases this may even lead to the inability to converge at all.

This issue is exacerbated in weight-sharing architectures such as CNNs, given that it is more difficult to control the exact output of multiple neurons given a single weight change. This can be observed in the following experiment where we simply employ the shallow CNN in combination with an unaltered version of FF (VFF/s). Figure 4 clearly shows that the accuracy declines upon going deeper into the network, which is a large issue in many cases.

Finally, due to the abovementioned issues, finding a correct value for $\tau$ is complicated. In limited experiments, we found the value that achieves the highest accuracy to be highly dependent on the architecture and dataset. However, all statements made or observed trends for a given $\tau$ (such as Figure 4) are verified to hold within the range [1;10].

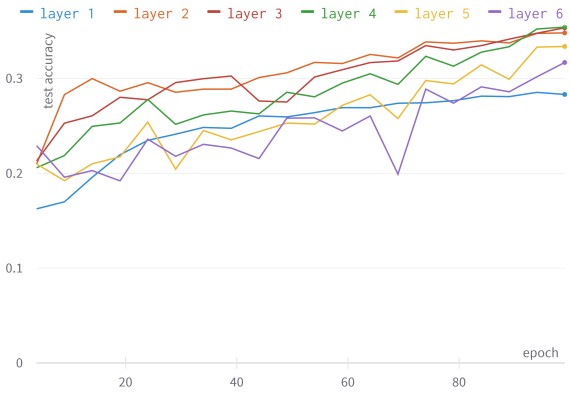

Figure 4: Layerwise accuracy of the best-performing VFF/s using the original loss function ($\tau = 2$) (Table 1). The experiment is performed on CIFAR-10.

## D  Exploring the New Loss Function

This section of the appendix discusses the properties of the new loss function and their impact on the accuracy and stability of Forward-Forward. We conclude by outlining some alternatives and extensions to our approach. For convenience, the new loss function is repeated.

$$L_{SB}(g^{pos}, g^{neg}) \triangleq \xi(g^{pos} - g^{neg}) \tag{9}$$

This function suffers from a flaw that makes it possible to lower the loss function without learning new features. As the loss is simply calculated by the separation, scaling all parameters uniformly (given an already existing separation) will lower the loss, not by learning but by simply amplifying the features. Mathematically, we can note the following.

$$\delta = ReLU(F_i(x, \theta_i)) \implies \alpha \cdot \delta = ReLU(F_i(x, \alpha \cdot \theta_i)) \tag{10}$$

This simply denotes that if the parameters of a layer $\theta_i$ are able to achieve a separation $\delta$, an arbitrary separation can be achieved by choosing $\alpha$ accordingly. Notice this only holds, if $F_i$ is a linear operation, which is the case for a convolutional or a dense layer. Moreover, ReLU only disregards negative features which has no impact on scaling the positive features (like sigmoid would).

In theory, this uncontrolled scaling can have devastating effects on the stability of a network. However, empirically, we find that this behavior is negligible in all experiments performed. We verify this using the three experiments listed below. To ensure the conjectured behavior has the highest chance to be observed, we perform these experiments on the model that achieves the highest accuracy on CIFAR-10, shown in Figure 3. This model has the most layers and is trained the longest.

**Plot the Mean of Positive and Negative Goodness Throughout Training for Each Layer.** This experiment provides a comprehensive view of the evolution of the goodness which can be used to indirectly assess the stability of specific layers through training. Figure 5 shows that some layers have significantly higher goodness than others, especially the second layer. Regardless, knowing that this is the squared sum of thousands of features, all values remain within the expected range for neural networks and do not demonstrate this uncontrolled scaling.

**Measure the Maximal Weight for Each Layer After Training.**  The magnitude of weights is only an indirect indication of stability or other common metrics. Nevertheless, low maximal weight ensures that the network suffers less from phenomena such as overfitting or instability. This experiment, depicted in Figure 6, reveals that the highest weight has a value of 0.3 which occurs in the first layer. After the second

layer, there is no layer with a weight larger than 0.2. After more epochs, there is no significant difference in maximal weight. This further refutes the notion that the network is simply scaling its weights to achieve higher separation.

**Measure the Maximal Goodness Value at Test Time.** The first experiment examined the mean values of goodness at training time to assess the stability of the network. However, simply examining the mean does not provide information about outliers, which may exclusively cause the issue. Comparably, examining training samples is inadequate as the model may exclusively become unstable outside of the training distribution. Therefore, this experiment studies the 'worst case' of the model; the maximal goodness at test time. We measure a maximal goodness of about 160 in the first three layers, then regressing to a maximum of 90 in the later layers, depicted in Figure 6. This, again, is within the realm of reason. Generally, longer training times do not severely impact the maximal activations except for the last layer. This effect can be seen to a lesser extent in Figure 5 (green line). We suspect this phenomenon is specific to this exact architecture (as it was not observed elsewhere) and therefore do not draw conclusions about its origins. Lastly, the peak at the second layer seems to be simply incidental and does not occur for different networks/datasets.

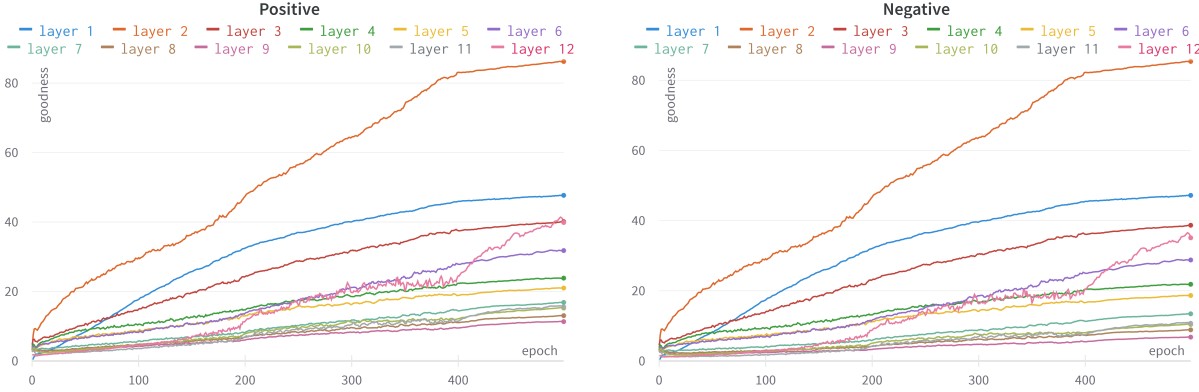

Figure 5: The evolution of positive (left) and negative (right) goodness during training for the best-performing CIFAR-10 TFF/d model presented in Table 2.

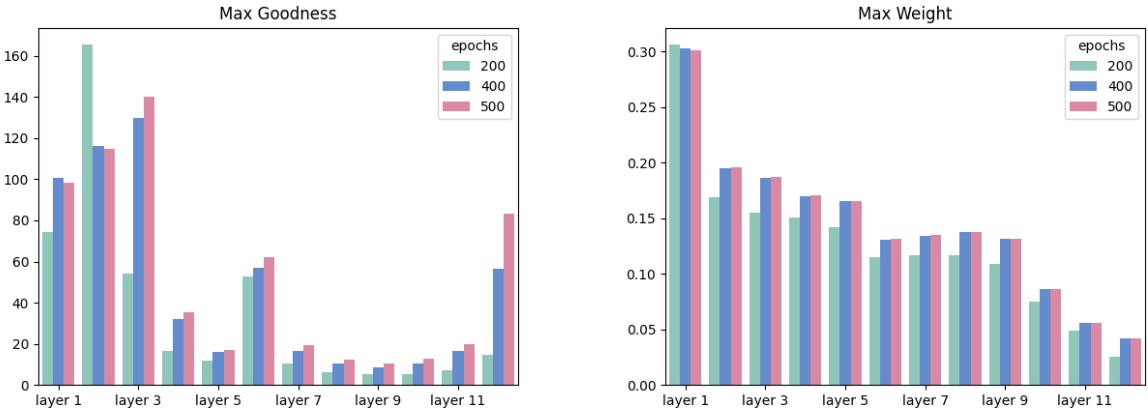

Figure 6: Layerwise maximal goodness (left) and maximal weight (right) at three different epochs for the best-performing CIFAR-10 TFF/d model presented in Table 2.

**Consequences.** We hypothesize that the explosion of weights practically does not occur due to variance in the network, known as Internal Covariate Shift (ICS) (Ioffe & Szegedy, 2015). In effect, it seems to be better to learn robust features instead of scaling highly specific ones to achieve the best separation. As the first two layers are barely subject to such variance, the optimization is more prone to increase their weights and therefore goodness. However, no further experimentation is performed to fully back up this hypothesis.

The only adaptation this work has made to avoid issues related to high values is the method of calculating the loss itself. Namely, the term $exp(g^{pos} - g^{neg})$ can lead to numeric instability, especially when working with half-width floating point values. Therefore, we approximate $log(1 + exp(g^{pos} - g^{neg}))$ with simply $g^{pos} - g^{neg}$ after a specific point [8].

**Alternatives.** A possible remedy is to find a loss function $L$ that satisfies the condition $\forall \alpha : L(pos, neg) = L(\alpha \cdot pos, \alpha \cdot neg)$. If the loss remained identical regardless of any $\alpha$, there would no longer be an incentive to scale up all parameters. However, consider the case where $\alpha = 0$, this would imply $\forall x, y : L(0, 0) = L(x, y)$ which only a constant function satisfies. Consequently, it is not feasible to achieve such behavior given the current approach.

Alternatively, a possible constraint can be put on the growth of weights by imposing weight decay. Limited experiments were performed with decay values ranging from $10^{-4}$ to $10^{-6}$ without success. In all cases, accuracy was significantly lower.

## E Normalization Deep Dive

This appendix accompanies section 4 and provides additional plots and insights about the impact of normalization. Specifically, the statements made in that section will be substantiated further here.

In contrast to the majority of other experiments in this work, this section uses the FCN architecture consisting of 6 layers that all have a hidden dimension of 2048. This choice is made to ensure that any trends throughout layers are the product of the learning algorithm and not the architecture (for example side effects of a maxpool). The observations made in this section are also verified to hold for CNNs, but not shown.

**No Normalization Leads to Feature Recycling.** We empirically demonstrate the feature recycling phenomenon as denoted in (Hinton, 2022). The recycling behavior stems from the reuse of features that already produce high/low goodness depending on the sample. Any subsequent layer can therefore simply perform an identity operation or even a permutation and produce low loss without learning anything new.

The following plots (Figure 7) demonstrate that networks trained without normalization, have substantially lower gradients, especially after a few layers. Compared to the other normalization strategies, the right-hand side of Figure 7 shows that the gradients of the network without normalization are several orders of magnitude lower. Further, the left-hand side shows that the magnitude of the gradients decreases throughout layers.

**Layer Normalization Leads to Information Loss.** Information loss is often quantified in terms of mutual information. Specifically, if an embedding has low mutual information with the input, it is said to have lost information. Informally, mutual information conveys how much information is shared between two distributions. Or alternatively, how much information does sampling one distribution reveal about the other and vice-versa. Consider a very narrow (bottleneck) layer that only has very few features, generally, it will be impossible to redetermine the original input exactly. However, this phenomenon occurs not only when reducing dimensionality but at many locations. For instance: a ReLU operation loses all negative information, and numeric approximation (which occurs at almost every floating point computation) loses information too.

The mutual information between the input and the label is generally low, from the label alone, the exact input cannot be determined. Therefore, at some point, the network will lose information. It has been shown that modern networks retain most information until the classification head aggressively narrows dimensionality (Shwartz-Ziv & Tishby, 2017). This allows later layers to manipulate data to become even more separable

---

[8]The PyTorch SoftPlus function has this functionality built-in by default with a threshold of 20.

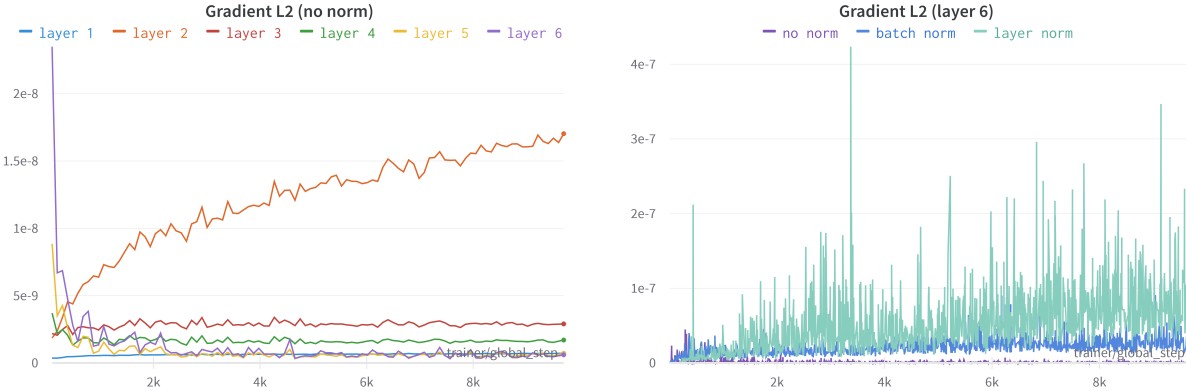

Figure 7: Layerwise gradient $\ell_2$ norms without normalization on the FCN network (left) and $\ell_2$ norm of the gradients of layer 6 compared to other normalization schemes (right). Both plots are on CIFAR-10 with the VFF+SymBa/f model. (The right figure is the same as Figure 2 supplemented with the no normalization model.)

while retaining a holistic picture of the input. In ordinary neural networks, mutual information is not essential to consider. There is simply one objective and the entire update is to better satisfy this objective. In this context, it is in the best interest of lower layers to retain information. In the case of FF however, each local objective has no incentive to retain information. Therefore, we need to ensure information loss is diminished.

However, this paper does not focus on the mathematical properties and behavior bounds but rather takes an empirical approach. We measure the usefulness of representations by appending a multi-layer classifier on top of each layer separately (Figure 8). The original network is pre-trained with VFF+SymBa/f and the classifier heads are trained with backpropagation for 100 epochs. Intuitively, we measure how a well-tested architecture and learning method can use the produced features by each layer for classification. This reveals that normalization affects the downstream usefulness of representations quite strongly. Concretely, as depth increases, the usefulness of layer normalization features quickly decays while no normalization retains them best, batch normalization stays in the middle. This relates back to finding the balance between forcing the network to learn new features (layer normalization) and simply reusing features (no normalization).

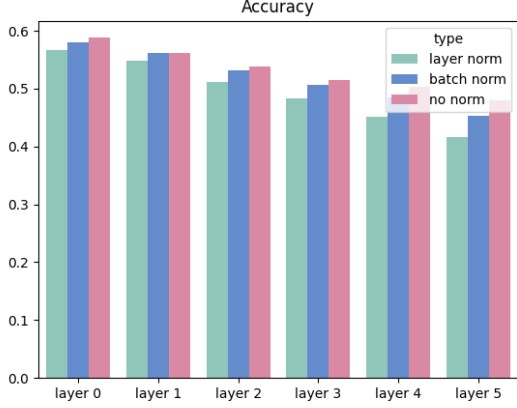

Figure 8: Layerwise test accuracy of auxiliary backpropagated classifier for layernorm, batchnorm, and no normalization. The experiment is performed on the VFF+SymBa/f model after 100 epochs on CIFAR-10.

**Batch Normalization Leads to Lower Variance.** Figure 2 depicts the $\ell_2$ norm of the gradients in each batch. The plots clearly indicate the increased variance of layer normalization in comparison to batch normalization. As additional evidence, it is possible to examine the weights of the models themselves. There are a multitude of techniques to visualize the distribution of weights. We opt to measure the maximal weight at each layer and the maximal goodness for each layer (on the full test set), depicted respectively in the left and right plots of Figure 9. This reveals that the network with layer normalization has higher maximal weights at each layer. Additionally, while less noticeable, the maximal goodness values are more constant throughout the batchnorm network.

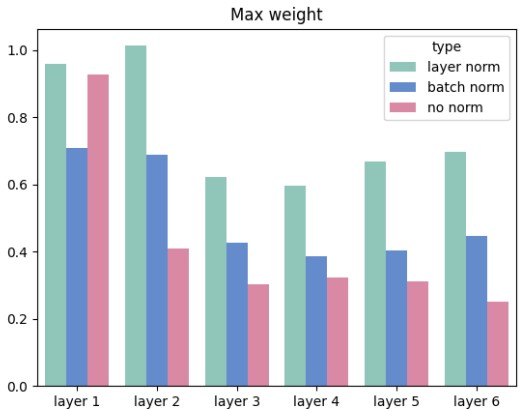 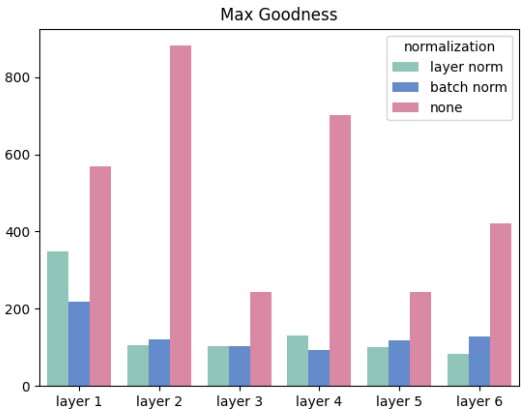

Figure 9: Layerwise maximal weight (left) and maximal goodness (right) for layernorm, batchnorm, and no normalization. The metrics are from the same run as depicted in Figure 8.

Maximal values are good for putting bounds on the variance of a network but give a narrow view of the distribution. Simply looking at the weights themselves (Figure 10) gives a qualitative insight into the differences of distribution: batchnorm has drastically less spread out weights.

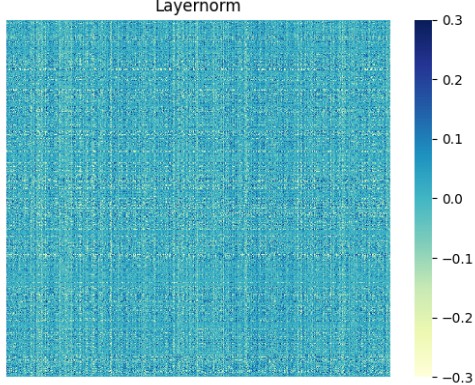 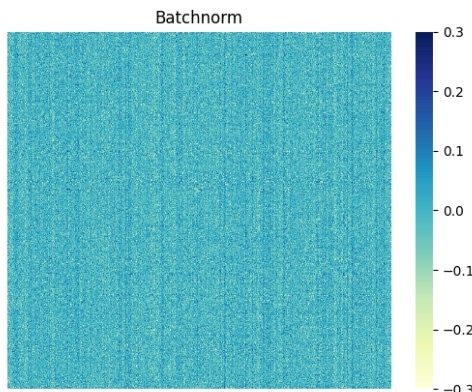

Figure 10: A visual representation of the weights at layer 6 of the VFF+SymBa/f model. The y-axis corresponds to the output neurons and the x-axis to the input neurons. Additionally, the values are clamped within [-0.3; 0.3] for visual clarity. The metrics are from the same run as depicted in Figure 8.

**Single Sample Training.** A notable advantage of layer normalization over batch normalization lies in the fact that there is no constraint on batch sizes. In a biological setting, which FF endeavors to resemble, learning by normalizing over a batch of training samples is implausible. However, comparable to evaluation using batch normalization, this can be solved by using rolling averages for the full training process. Some preliminary experiments have shown that this does not significantly degrade accuracy. Specifically, when using the same settings as in Table 1 with the full TFF, the resulting model achieves an accuracy of 73.3%, only 2% lower than originally.

## F    Error Signals and Overlapping Updates

This appendix will provide an overview of the role and implementation of Overlapping Local Updates (OLU) within FF. Prior to this, a detailed explanation of OLU is given.

**Details of OLU.** Intuitively, OLU is a form of truncated backpropagation where two layers are updated at once. At each step in the learning process, instead of simply updating the current layer given the local objective function, the previous layer is also updated. In the context of FF, there are two possible implementations. Both approaches are depicted in Figure 11.

- Approach 1: Alternatingly update a group of 2 layers each epoch (used in this work).

- Approach 2: Update both the current and previous layer at each step.

In terms of accuracy, both techniques are equal. The discrepancy lies in their runtime; we found the first approach to be faster than the second on the setup we used. Therefore, we opted for the first approach. However, we expect, with some tuning, both techniques can perform equally fast.

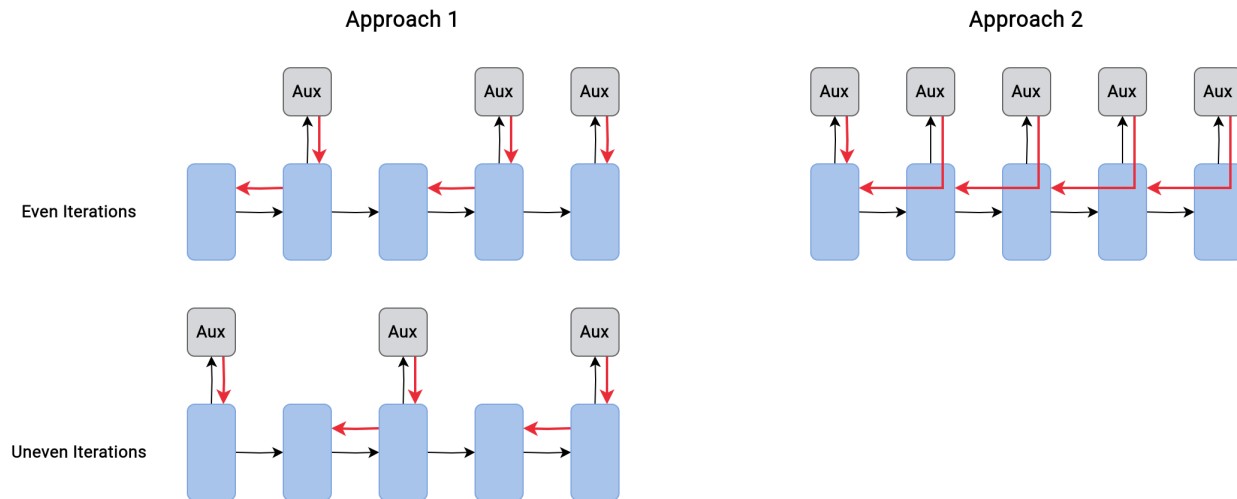

Figure 11: A visual representation of two possible OLU implementations. The red lines represent error signals. At even iterations, even objectives are optimized and at uneven iterations, uneven objectives are optimized.

**Alternatives to OLU.** In the case of this work, OLU was inspired by the concept that neurons not only send signals to other neurons within their layer but also to neighboring neurons, such as the previous layer. OLU takes the simplest approach from a contemporary research standpoint and allows a full error signal to be locally propagated. However, more biologically plausible alternatives exist, in which random synaptic feedback can be used to locally update neurons (Lillicrap et al., 2016). We encourage future work in this area to explore this technique.

## G   Architecture Discussion

The main role of this appendix is to depict and describe the two main architectures used throughout this work. However, before that, a collection of empirical observations will be provided that led us toward this design. These observations will be contrasted with backpropagation and outline the differences and similarities to this algorithm.

During the experimentation phase of this work, the viability of training several architectures with FF was tested. This manual search focussed on simple CNNs, meaning only combining convolutions and maxpool operations. This exploration entails more 'hyperparameters' than might be expected. For instance, technically FF is proposed as a layer-wise learning algorithm. However, it can be used as a general algorithm to train entire 'blocks', such as (He et al., 2016; Szegedy et al., 2015), or simply a subnetwork. However, this work limits itself to layerwise training and defers this avenue of research to future work.

When training simple CNNs, two parameters of the network are of importance: the width and the depth. Varying these parameters directly influences the learning speed (time to reach a threshold in accuracy) as well as the final accuracy (accuracy convergence after long training). The learning speed difference between shallow and deep networks is a well-known phenomenon in backpropagation (especially in non-residual architectures). This stems from the increasing instability of gradients (He et al., 2016). In local learning, as no global gradients are used, this issue does not arise.

The accuracy of the local algorithms is mostly determined by the width (the number of features or channels) of the layers. There exist several architectures that intentionally reduce the number of features of certain layers, such as autoencoders (Rumelhart & McClelland, 1987). These networks can be trained with backpropagation which will result in these narrow layers containing a compressed representation of the input. However, in local learning regimes, this bottleneck simply leads to information loss as the local objective struggles to retain useful downstream information. Conversely, scaling the width of a layer severely impacts the learning speed, intuitively, more features lead to a more challenging optimization process as local minima are harder to find. Therefore, the dimensionality of the features after each block is of paramount importance to the final accuracy.

When using local learning on CNNs, this balance in network width can be balanced through lossy operations such as maxpool operations. Therefore, their locations severely impact the learning characteristics of the network. This disparity causes a large difference between the shallow and deep networks used in this work. As can be seen on Figure 12, the shallow network is not simply a subset of the deep network. Empirically, as discussed in section 6, we confirm the expected tradeoff between learning speed and final accuracy between these architectures.

As explained in section 5, each layer of our architecture deviates from the ordinary order of operations. Particularly, each layer first normalizes, followed by the convolution and the non-linearity. Lastly, an optional max pool is performed before the calculation of the goodness, which is explicitly shown in Figure 12. This figure shows the progression of channels through the architecture, where $C$ is the number of input channels.

## H   Evaluation Strategies

As originally discussed in section 3, there are several methods to evaluate with a network trained in a supervised FF setting. This appendix explores some such methods and compares their effect on the final accuracy.

**Goodness-Based Classification.** This evaluation method stands out as the prominent classification technique to achieve the highest possible accuracy with FF (Hinton, 2022). This stems from the fact that this technique leverages the exact purpose the network was trained for. It evaluates the unseen example with all possible labels in the dataset and selects the label with the highest goodness, as this is the most positive. The main disadvantage, however, is that it requires an evaluation of the network for each possible label, which is prohibitively expensive in most cases.

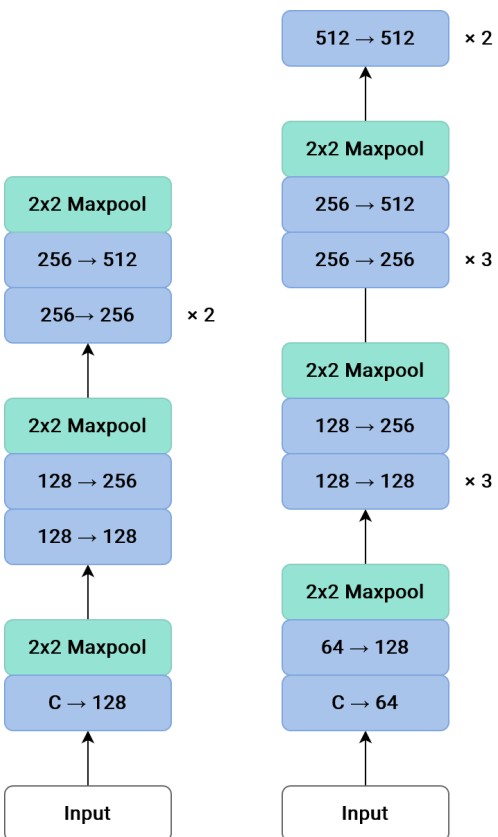

Figure 12: The two CNN architectures used throughout this work. The shallow network (left) has 2.8 million parameters and the deep network (right) has 8.5 million parameters.

**Feature-Based Classification.** This method does not necessitate sampling each class individually. Instead, it utilizes a neutral label, which is the average of all class labels, to generate features that represent all classes. These features are then used by an auxiliary network to make a single prediction encompassing all classes. In line with (Hinton, 2022), we find that this method generally suffers from instability and therefore does not achieve the accuracy of the aforementioned technique. We have performed several experiments to remedy this without success: freeze the whole network except the auxiliary head, use normal negative labels instead of neutral ones, use very low learning rates, and use a classifier consisting of multiple layers.

**Ensemble Classification.** The fact that the objective of each layer is the same can be leveraged in both aforementioned techniques. Unlike ordinary backpropagated networks, which only use the last layer to generate a prediction, any layer or combination of layers of the network can be used for evaluation. In essence, the goodness or features from multiple layers can be aggregated into an ensemble that is more resilient to noise and generally achieves higher accuracy. This aggregation can be a straightforward average or even a small classifier of any kind. To predict, the highest value of the mean goodness is used, instead of a single goodness value, which results in more stability. This is depicted in Figure 13. In this work, we use the average of the goodness to make the prediction. With this technique, we are able to augment the test accuracy of all our deep models by a few decimals, shown in Table 3. The shallow models do not benefit from averaging the last few layers as the accuracy of the last layer is significantly higher than its predecessors.

## I  Residuals

Residual connections He et al. (2016) have proven vital to train deeper networks with backpropagation. Therefore, utilizing them to improve FF seems like a clear path to improve the scaling behavior. The

Table 3: Comparison of two proposed evaluation strategies on the test accuracy. The shown models are the same as the TFF/d models at 500 epochs from Table 2.

| evaluation | MNIST | F-MNIST | SVHN | CIFAR-10 |
|---|---|---|---|---|
| only last layer | $99.58 \pm 0.06$ | $91.38 \pm 0.38$ | $94.31 \pm 0.07$ | $83.51 \pm 0.78$ |
| three last layers | $99.62 \pm 0.02$ | $91.51 \pm 0.27$ | $94.36 \pm 0.04$ | $83.72 \pm 0.80$ |

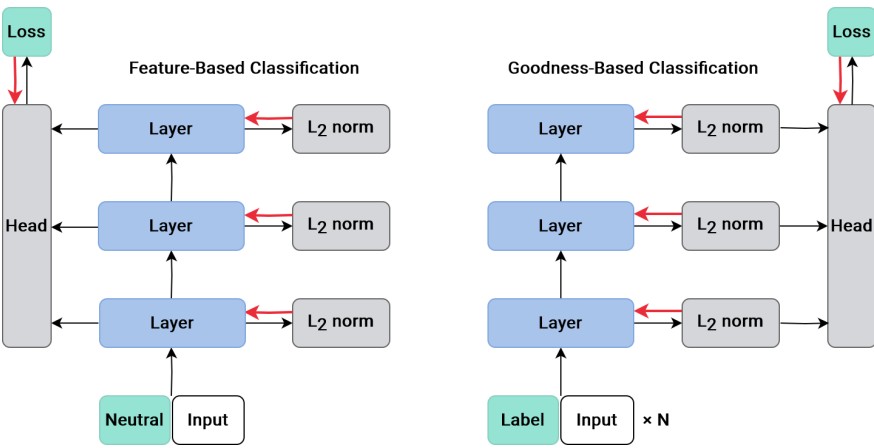

Figure 13: A depiction of the two prominent evaluation techniques for Forward-Forward. The black arrow represents data flow and the red arrows represent gradient flow. Goodness-based classification (right) requires the network to be evaluated $N$ times according to the amount of label. The 'head' in this figure can range from a simple average to a small classifier.

motivation between residual connections is twofold: it provides a sensible (indirect) initialization for each layer, namely the identity matrix, and allows the backward gradients to bypass each layer. This results in a smoother convergence, especially for deeper networks.

Within FF, both motivations no longer hold. First, adding the identity to each layer is not desirable as this may discourage the network from learning new features. Second, there is no need for backward gradients to bypass layers, as there is no full backpropagation. Additionally, the models used within this work are too shallow enough to fully benefit from residuals, even if they were trained with backprop. Interestingly, preliminary experiments using TFF/s and TFF/d on CIFAR-10 with residual connections paint a more neutral picture. Specifically, early during training (the first 30 epochs), the use of residual connection yields improved results. However, after 100 epochs the accuracy of the small and deep models are respectively 4% and 2% lower. Therefore, we believe residual connections to hold some promise, if modified in a sensible manner to the requirements of FF.

