# OpenReview forum: "The Trifecta: Three simple techniques for training deeper Forward-Forward networks"
_TMLR — Accepted by TMLR_

### Review · Reviewer_j1gm · 2024-08-25

**Summary Of Contributions:**

This work examines the issues of the recently proposed Forward-Forward learning algorithm and suggests certain modifications to the original training procedure, including the choice of the loss function, feature normalization, and update strategy. These modifications significantly enhance the performance of the algorithm. The proposed Trifecta prescription is validated on several image classification datasets and CNN/MLP architectures.

**Audience:**

Yes

**Claims And Evidence:**

Yes

**Requested Changes:**

- In my opinion, the background on the forward-forward algorithm should be slightly expanded for readers not familiar with the topic. While the loss function is mentioned, the exact update rule (optimizer step) is not provided. I suggest adding this for the sake of completeness.

- Whereas the extension of the approach to residual networks is left for future work, is there a significant challenge with the application of forward-forward to small residual networks? There are small 9-layer ResNets [1] that can be rapidly trained on the CIFAR-10 dataset to achieve pretty good accuracy. It would be interesting to examine how the FF algorithm works with residual connections.

- [Minor] I think it would be better to have a sequential color scheme for the plots in Figures 1 and 3, such that a darker/brighter tone (or any other rule) indicates deeper layers, making them easier to discern.

[1] https://github.com/apple/ml-cifar-10-faster

**Strengths And Weaknesses:**

*Strengths*

- The problem under study is of significant importance given the rapidly increasing scale of neural networks and the demand for memory-efficient alternatives to the backpropagation algorithm.

- The paper is clearly written and well-structured, providing both theoretical intuition and experimental evidence to support the importance of each component of Trifecta.

- The proposed Trifecta methodology significantly advances the performance of the forward-forward learning algorithm compared to prior work on several datasets. Trifecta not only improves performance but also accelerates convergence.

---

> ### Author Response · Authors · 2024-09-09
>
> We thank the reviewer for their positive comments and useful feedback. We are also happy that the reviewer appreciated the clarity of your work.
>
> **Slightly expand Background**
>
> We have updated the introduction, introducing the original algorithm further, especially aimed at readers less familiar with the topic. We also added the update step.
>
> **Examine how the FF algorithm works with residual connections.**
>
> Regarding residuals; implementation-wise, this is not challenging to add. Appendix J briefly covers some positive experiments by naively added these connections. However, we are conceptually unsure whether residuals are the best fit for FF (which is also briefly covered in that appendix). Concretely, the original motivation for residuals make less sense in the local learning regime since the learning algorithm can't fully leverage it. On the other hand, new inductive biases, keeping into account these changes, are quite likely to exist. We did not feel like we could do the topic justice in this paper, both empirically and intuitively. We therefore decided not to include it as fourth component to the Trifecta.
>
> **Use a sequential color scheme for the plots**
>
> We changed the plots to have a more natural color progression.

---

### Review · Reviewer_R5wW · 2024-09-01

**Summary Of Contributions:**

This paper highlights three weaknesses of the original Forward-Forward (FF) algorithm. These include, the loss function, the normalization function and the lack of error signal. The paper then proposes three simple techniques that improves the performance of the FF algorithm on deeper neural networks. They show that their techniques allow FF to reach training speeds and test accuracy that is on par with traditional backpropogation on simple datasets.

**Audience:**

Yes

**Broader Impact Concerns:**

None.

**Claims And Evidence:**

Yes

**Requested Changes:**

Requested changes listed above.

**Strengths And Weaknesses:**

### Strengths
- Discovering the positive synergy between these three components can give insights into how we should combine different techniques used in the FF algo.
- The paper provides an thorough related works section and does a good job of placing it within the larger body of literature.
- The results are quite impressive for datasets such as CIFAR-10.

### Weaknesses and Requested Changes

- I wouldn't really consider SymBa a contribution of this paper. I suggest removing SymBa as part of the Trifecta as it was already proposed by Lee & Song (2023).
- Many details about how Batch Normalization and Overlapping Local Updates helps are left to the Appendix. These explanations should be in the main body if they're part of the core contributions.
- The plots in Figure 2 would benefit from some smoothing of the curves so that we can see the general trend.
- I am not totally familiar with the literature on local-learning but I am sure you could train the network to have high/low goodness at each layer by using backpropagation as well. Say by setting the loss as the average of the separation gap across all layers? How would this compare in terms of test accuracy to the current Forward-Forward implementations?

---

> ### Author Response · Authors · 2024-09-09
>
> Thanks for the clear feedback. We appreciate it.
>
> **I suggest removing SymBa as part of the Trifecta**
>
> We respectfully disagree with this. The goal of the paper is to fix the apparent weaknesses of the FF algorithm. The motivation is twofold; show that local learning can be competitive on small/medium datasets and providing a new starting point for further research. We found these three methods to synergise very well together. We believe removing SymBa from this trifecta would result in more confusion. Lastly, we never claim that this technique is ours, we merely combine and enhance it.
>
> **Details are left to the Appendix**
>
> The explanations of the mentioned topics span several pages. We felt that this was too much information to put in the main body (even though it is part of our contributions). We believe to have struck a right balance between conciseness and rigour however if the reviewer has any specific explanation or figure they would like to see moved to the main text, please let us know.
>
> **Figure 2 would benefit from some smoothing**
>
> Figure 2 is meant to show how much the gradients vary throughout steps, which is significantly higher when using layernorm, indicating unstable optimization. The means are roughly the same across algorithms so smoothing would lose the important information the plot aims to convey. However, we appreciate the feedback that the current plot does not convey this well. We therefore updated this plot toward a trendline + variance depiction. We are happy with the result and thank the reviewer for this idea.
>
> **Train the network to have high/low goodness at each layer by using backpropagation**
>
> In a sense, this is an extended version of OLU, where gradients are allowed to flow across the whole network instead of simply one. While interesting, this defeats the point of local learning (only using local information to compute weight updates), which is the main point of using FF.
>
> We happily invite more feedback on our comments.

---

> > ### Comment · Reviewer_R5wW · 2024-09-13
> > **Follow up**
> >
> > I thank the authors for their response.
> >
> > I am still not really satisfied with the use of SymBa as part of your contributions. Unlike BatchNorm and OLU which have not been investigated in the context of FF, SymBa was originally designed to address the poor loss function in the original FF algorithm. The fact that it is useful for FF, and why it is useful, was already known. In what way are you enhancing SymBa?
> >
> > To me it seems that the authors have just shown that SymBa does not counteract the benefits from BatchNorm and OLU.

---

> > > ### Author Response · Authors · 2024-09-15
> > >
> > > Thanks for the response. We wanted to provide some arguments behind our reasoning for including it.
> > >
> > > - The inclusion of other work into a proposed solution has been done in many well received papers in the past (e.g. https://arxiv.org/pdf/1710.02298) and is common scientific practice.
> > > - We think the fact that SymBa not only works together with the other techniques but results in an accuracy improvement that is greater than the sum of its parts a worthwhile contribution and a reason to include it in the Trifecta (which is what me meant by combine *and enhance*).
> > > - We've taken care to properly attribute the SymBa work throughout our paper. Each paragraph discussing SymBa begins by acknowledging their contributions, and we explicitly state that they, not us, solved the problem in question.
> > > - Lastly, the fact that SymBa is included into the trifecta, and not merely mentioned in passing, improves attribution in our view.
> > >
> > > Consequently, we are currently not convinced that removing SymBa will present a meaningful improvement to this work. We're open to further discussion on how to improve our attributions to this previous work.

---

### Review · Reviewer_YJVK · 2024-09-05

**Summary Of Contributions:**

The goal of the paper is to study alternatives to backpropagation for training neural networks. First, the paper recalls a recent algorithm, the Forward Forward algorithm (Hinton 2022), and highlights three of its issues: the loss function, the normalization, and the potential lack of signal. The authors propose to fix each of these 3 issues, hence the ´trifecta´ coining.

**Audience:**

Yes

**Claims And Evidence:**

Yes

**Requested Changes:**

- clarification on  ´backpropagation is overfitting´
- more detail on what is done in the Forward Forward algorithm, and the proposed algorithm: add precise equations and algorithms
- comments on the difference with the paper "Can forward gradient match backpropagation?."

**Strengths And Weaknesses:**

Strength: The tackled problem is hard, important, and interesting, especially with potential application to large language models, for which one cannot resort to vanilla backpropagation. To the best of my knowledge, making this type of backpropagation-free algorithm work in practice can be very challenging, and this paper provides tricks to obtain performances significantly better than previous backpropagation-free algorithms.



Weaknesses:
Intro: - To the best of my knowledge, backpropagation is a way of computing gradients, using the chain rule and computing the derivatives in an efficient reverse order. IMO it is weird to say ´backpropagation is overfitting´ since it is a way to compute the gradient, the exact gradient computed with the forward mode would yield the exact same result as backpropagation.


Background:
- IMO more details are required when introducing the Forward Forward algorithm, maybe recall the algorithm?
- More generally I can understand the ideas of the paper, which are very well explained with words, but it feels like the paper would benefit from more concrete ¨maths¨, e.g., more formal equations and algorithms. I would love to see the exact updates of the algorithm and a precise comparison of the updates.


Literature Review: Could the authors contextualize their work compared to [1], where good performances on CIFAR are reported with local updates?

[1] Fournier, Louis, et al. "Can forward gradient match backpropagation?." International Conference on Machine Learning. PMLR, 2023.




Minor:
- Harmonize bibliography style: chose FirstName LastName or F. LastName, e.g. Y. Bengio should be Yoshua Bengio in Appendix
- ~~I would recommend using matplotlib for the figures instead of wandb output. Very very minor: in particular I would recommend using the viridis colormap for readability of the indices of the layers~~

---

> ### Author Response · Authors · 2024-09-09
>
> We thank the reviewer for their thorough feedback.
>
> **It is weird to say ´backpropagation is overfitting´**
>
> We agree, we were originally somewhat colloquial in the first sentences to make the challenges clear. We revised manuscript to be more accurate in the proposed regard.
>
> **More details are required when introducing the Forward Forward algorithm, maybe recall the algorithm?**
>
> This is very fair and has been suggested by other reviewers. We expanded the background section to include more general intuitions as well as concrete algorithmic steps.
>
> **The paper would benefit from more concrete ¨maths¨**
>
> We generally dislike papers that only use formulae for the sake of maths. We feel like we disambiguate where necessary but without overwhelming new readers. Therefore, the appendix contains much more algorithms and maths for the interested reader. However, we added and highlighted the formulae in the body. Is there any additional formula or passage that the reviewer thinks would benefit from more rigour?
>
> **Could the authors contextualize their work compared to [1]**
>
> Thanks for the pointer, we didn't come across this in our literature review but we discuss the comparison. This has not yet been added in the manuscript at the time of writing but will do so as soon as possible.
>
> **Harmonize bibliography style**
>
> We are unsure what the reviewer means. We simply used the standard TMLR format and use citet/citep as is generally done.
>
> **Update figures to be more clear & professional**
>
> We used plotly to produce my figures and employed more readable and clear colorschemes.
>
> Let us know if there are any additional points of feedback.

---

> > ### Comment · Reviewer_YJVK · 2024-09-10
> >
> > > We generally dislike papers that only use formulae for the sake of maths. We feel like we disambiguate where necessary but without overwhelming new readers. Therefore, the appendix contains much more algorithms and maths for the interested reader. However, we added and highlighted the formulae in the body.
> >
> > I agree that too much can quickly become overwhelming. However, it feels that the paper is currently not self-contained: I do not think I would be able to reimplement it myself with the current details.
> >
> > > Is there any additional formula or passage that the reviewer thinks would benefit from more rigour?
> >
> > Could you recall the formulae of $g_{pos}$ and $g_{neg}$? (it feels this is important to understand the weight update formula)

---

> > > ### Author Response · Authors · 2024-09-10
> > >
> > > Thanks for the quick response.
> > >
> > > $g^{pos}$ and $g^{neg}$ can be computed in several ways. The original paper proposes two methods, supervised and unsupervised. The former (supervised) encodes the label in the image using a one-hot vector. We use a similar technique but instead encode the label as an extra channel (this is explained in the "training and encoding" section). The latter (unsupervised) generates negative samples by merging samples into strange (but somewhat spatially coherent) images.
> > >
> > > Since there isn't a set way to compute these samples, we didn't go into detail in the background section. However, in an upcoming revision, we added formulae on how we append our labels to positive and negative samples.
> > >
> > > In there anything else that the reviewer would like to see?

---

### Decision · Action_Editor_v2xi · 2024-10-09

**Recommendation:** Accept with minor revision

**Comment:**

All reviewers agree that the claims are mostly well-supported and there is strong empirical evidence for their proposed method.

However, 2/3 of the reviewers feel that the forward-forward (FF) algorithm, on which the current submission builds heavily on, is not properly described and lacks clarity. The authors did provide a little more background but I agree that the FF algorithm needs much more detail than what is provided. Currently there is less than a page devoted to this algorithm, which feels inadequate given the centrality to understanding, appreciating, and reproducing this work. If the authors are concerned with space, at a minimum I would suggest adding a lot more detail to the appendix.

Given that all reviewers would be supportive of an acceptance as long as more details on FF are provided, I am recommending an acceptance with minor revisions, where it should be made clear to the authors that I will only accept the camera-ready version as long as substantially more details are provided for FF.

**Audience:**

This paper would be of interest to a wide audience, provided the authors provide sufficient background, in particular with respect to the forward-forward algorithm on which their work builds on.

**Claims And Evidence:**

All reviewers agree that the claims are mostly well-supported and there is strong empirical evidence for their proposed method.

**Resubmission Of Major Revision:**

The authors may consider submitting a major revision at a later time.

---

> ### Author Response · Authors · 2024-10-20
>
> We thank the reviewers and the AE. We sincerely believe this review process has resulted in a more refined and complete manuscript.  As for the final revision, since the original manuscript, we have made the following changes:
>
> We fully revised the first half of the background section (Sec. 3) which discusses the Forward-Forward Algorithm.  This was originally somewhat unclear due to brevity constraints.
> - For instance, we now added a new subsection (Sec. 3.1) to discuss the negative sampling and why that is necessary.
> - We also gave more context to the algorithm introduction, being much more specific.
> - We reorganized the section slightly for a better flow.
> - We further explained the weight updated and provided formulae (Sec 3.2).
> - Finally, we added pseudo-code to further clarify any remaining doubts.
> - Also, we contrast with this pseudo code in Sec 4 to clarify our changes.
>
> With the current changes, the description of the FF Algorithm provided in the paper is comparable to that in the original FF paper (Hinton, 2022).